# Double Strike in Chronic Lymphocytic Leukemia—The Combination of BTK and BCL2 Inhibitors in Actual and Future Clinical Practice

**DOI:** 10.3390/ijms26073193

**Published:** 2025-03-29

**Authors:** Przemyslaw Zygmunciak, Hanna Dancewicz, Katarzyna Stróżna, Olga Błażowska, Krzysztof Bieliński, Tadeusz Robak, Bartosz Puła

**Affiliations:** 1Faculty of Medicine, Warsaw Medical University, 02-091 Warsaw, Poland; zygmunciakprzemyslaw@gmail.com (P.Z.);; 2Department of Hematology, Medical University of Lodz, 93-510 Lodz, Poland; robaktad@csk.umed.lodz.pl; 3Department of General Hematology, Copernicus Memorial Hospital, 93-510 Lodz, Poland

**Keywords:** Bruton kinase inhibitors, B-cell lymphoma 2 inhibitors, chronic lymphocytic leukemia, therapy

## Abstract

In the recent 2024 ESMO guidelines, the combination of venetoclax and ibrutinib was listed as one of the first-line treatment options for CLL patients. These drugs were first-in-class medicines that revolutionized CLL management, extending patients’ overall survival even in cases refractory to immunochemotherapy. However, since the approval of both compounds, more and more Bruton Tyrosine Kinase inhibitors (BTKis) and B-cell lymphoma 2 inhibitors (BCL2is) have been discovered. Their efficacy and safety are the reasons for their use in monotherapy among both treatment-naïve and relapsed patients with CLL. Currently, several ongoing clinical trials are investigating the rationale for the combination of BCL2is and BTKis. In this review, we discuss the recent advancements in the field of co-therapy with BTKis and BCL2is.

## 1. Introduction

In the Surveillance, Epidemiology, and End Results (SEER) database, the age-adjusted incidence of chronic lymphocytic leukemia (CLL) is 4.9 per 100,000 population per year [1]. At the time of diagnosis, the median age is 70 [1]. Men are more often affected than women [2]. The incidence is expected to increase in coming years, as the disease is more prevalent in aging societies. Furthermore, with the rising use of diagnostics, CLL is more frequently diagnosed at earlier stages, even in younger patients [3].

For many years, the Binet and Rai classification scales served as the primary clinical classification system for patients, helping to determine the stage of progression, assess prognosis, and indicate treatment. However, both scales have become less useful with the introduction of novel effective treatments; they are no longer satisfactory for accurately determining patient prognosis. Additional markers are now being used to predict the prognosis of patients with CLL. Patients with *TP53* mutations or detectable del(17p) have the worst prognosis when treated with immunochemotherapy. Another unfavorable prognostic marker is unmutated immunoglobulin heavy-chain variable region gene (*IGHV*) status. It results in higher genetic instability with a higher risk of deleterious genetic mutations [4].

## 2. Treatment

In Binet A and B stages without active disease and Rai 0, I, and II without active disease, it is recommended that a ‘watch and wait’ strategy be adopted [4]. In the case of symptomatic disease or advanced-stage disease (Biner C or Rai III and IV), treatment-naïve (TN) CLL, various therapy options may be considered based on the presence of prognostic factors, e.g., *IGHV* mutational status, del(17p), and *TP53* mutation. Bruton’s tyrosine kinase inhibitors (BTKis) pose one of the treatment options, and include registered compounds, e.g., ibrutinib, acalabrutinib, and zanubrutinib, which are administered as continuous therapy until progressive disease or treatment intolerance occurs [5]. Alternatively, time-limited therapy with the B-cell lymphoma 2 inhibitor (BCL2i) venetoclax, in combination with obinutuzumab, may be utilized [5]. The ESMO guidelines also underline the possibility of the use of 3 cycles of ibrutinib monotherapy followed by 12 cycles of venetoclax and ibrutinib combination therapy. In cases where targeted therapies are not reimbursed and in patients with a favorable genetic risk profile, time-limited chemoimmunotherapy may be considered; however, considering fludarabine–cyclophosphamide–rituximab (FCR) toxicity, this treatment modality is currently regarded as obsolete [5]. Possible treatment options for early-stage CLL are shown in Figure 1.

The treatment for relapsed/refractory (R/R) disease depends on several factors, including the primary treatment option, comorbidities, access, *TP53* status, and/or del(17p) mutation status [5]. Generally, the patients who did not receive BCL2is as a first-line treatment would be assigned to either method, with the possibility of adding a monoclonal antibody to their regimen [5]. In the case of double-exposed or double-refractory patients, two possible treatment options have been registered: a non-covalent BTKi, pirtobrutinib (FDA and EMA approval), and lisocabtagene maraleucel (liso-cel; FDA approval) [6,7]. Considering the increasing vast use of novel agents, the number of double-exposed and -refractory cases has increased, and may pose a therapeutic challenge in the near future [8,9,10]. Such patients should qualify for clinical trials; however, several agents have already demonstrated promising efficacy as monotherapies or in combinations, such as BTK degraders, Chimeric Antigen Receptor-Positive T (CAR-T) therapy, NK (CAR-NK) cell therapy, and bispecific antibodies [11].

## 3. Bruton’s Tyrosine Kinase Inhibitors (BTKis)

Bruton’s tyrosine kinase is a crucial mediator in B-cell signaling cascades, facilitating the activation of essential survival pathways [12,13]. This signaling network is integral to B-cell receptor (BCR) signal transduction. The development of therapeutic agents targeting BTK has revolutionized the treatment landscape of B-cell malignancies, establishing these compounds as a highly effective class of targeted therapy [12,13]. Ibrutinib was the first approved BTKi and revolutionized the treatment of CLL. It was followed by acalabrutinib, zanubrutinib, tirabrutinib, and pirtobrutinib. All of these drugs, except pirtobrutinib, are covalent inhibitors of the active site of the enzyme [14].

### 3.1. Covalent BTKis

First-in-class BTKis were designed to covalently and irreversibly bind to cysteine-481 of the enzyme, leading to decreased phosphorylation of the molecule and downstream kinases [11]. Their binding site is important since the most commonly observed mutation among BTKi-treated patients is a cysteine-to-serine mutation (C481S), which reduces the efficacy of covalent BTKis [15]. However, ibrutinib and subsequent drugs, such as acalabrutinib and zanubrutinib, have become the first-line treatment for CLL due to their potent action and manageable adverse events (AEs). A comparison of BTKi features is displayed in Table 1.

#### 3.1.1. Ibrutinib

Ibrutinib is a small-molecule, irreversible BTKi responsible for inducing apoptosis in CLL cells and B-cell lymphomas [19]. A preliminary phase I study evaluated ibrutinib in 56 patients with relapsed or refractory B-cell lymphoma and CLL [13]. The study used two different dosing schedules—a 28-day on/7-day off regimen and continuous daily dosing. An analysis of 50 evaluable patients showed an objective response rate of 60%, including a complete response rate of 16%. The median progression-free survival (PFS) for the entire cohort was 13.6 months [19]. The phase III RESONATE trial further validated these efficacy findings, comparing ibrutinib treatment with six cycles of ofatumumab, an anti-CD20 monoclonal antibody [20]. Published long-term data with a median follow-up of 65.3 months showed that patients treated with ibrutinib had better PFS than patients treated with ofatumumab. The PFS at 3 years was 40% for ibrutinib and 3% for ofatumumab at the 60-month time point [21]. Recently, the data from ten years of observation were published and PFS in ibrutinib’s cohort was as high as 8.9 years, whereas median overall survival (OS) was not reached [22]. Moreover, patients harboring at least one high prognostic risk factor also showed a significant increase in PFS compared to those treated with chlorambucil. Of note, the overall response rate (ORR) and complete remission (CR/CRi) remain at the same level, reaching 91% and 36%, respectively.

A phase 2 trial conducted by Ahn and colleagues evaluated ibrutinib as an initial therapy in 34 CLL patients with *TP53* mutations [23]. Their findings showed that after 6 years of treatment, 61% of patients maintained PFS, while 79% were alive. Of the 12 patients who experienced disease progression while on ibrutinib, eight patients showed progression of CLL and four patients had Richter transformation. These outcomes suggest that single-agent BTKi therapy can effectively manage high-risk CLL with *TP53* abnormalities for extended periods in certain patients [23]. Despite its efficacy, the drug can cause a number of side effects, as has been shown in long-term studies, due to the inhibition of off-target kinases. Ibrutinib was shown to increase the risk of cardiac arrhythmias, especially atrial fibrillation, as well as bleeding, hypertension, and heart failure [24,25,26].

#### 3.1.2. Acalabrutinib

Acalabrutinib is also an irreversible, second-generation BTKi inhibitor; however, it is more selective than ibrutinib, improving the efficacy and safety of BTKi inhibitors [27]. In studies of R/R CLL patients treated with acalabrutinib, its efficacy, durability of response, and safety were confirmed. The initial evaluation occurred in a combined phase 1/2 trial involving 61 relapsed CLL patients. The phase 1 portion explored doses ranging from 100 to 400 mg daily, while the phase 2 expansion utilized 100 mg twice daily. The dose-escalation phase proceeded without any dose-limiting toxicities [28]. The subsequent expanded analysis encompassed 134 patients with R/R CLL/small lymphocytic lymphoma (SLL) who received acalabrutinib 100 mg twice daily for a median treatment duration of 41 months [29]. Most AEs were mild to moderate, primarily with diarrhea (52%) and headache (51%). Grade ≥ 3 events were less common, with neutropenia (14%) and pneumonia (11%) being the most frequent. Atrial fibrillation and major bleeding were rare, occurring in only 7% and 5% of patients, respectively [29]. Taking ibrutinib has been shown to increase the risk of atrial fibrillation [30]. The study was conducted on 24 patients who started treatment with ibrutinib. Within 12 months, four patients were diagnosed with atrial fibrillation [30]. Phase 2 studies of acalabrutinib were conducted in R/R CLL patients intolerant to ibrutinib [31]. Sixty patients were treated [31]. Of the 61 ibrutinib-related AEs associated with intolerance, 72% did not recur, and 13% recurred at a lower grade with acalabrutinib [31]. Three patients (5%) achieved CR, and the ORR to acalabrutinib was 73% [31]. A post hoc analysis of events of clinical interest (ECIs) further supports acalabrutinib over ibrutinib, showing a higher incidence of common AEs, such as diarrhea, arthralgia, and urinary tract infections, which were more pronounced in patients treated with ibrutinib [32]. Notably, the occurrence of atrial fibrillation/flutter, bleeding, and/or hypertension was more common in the ibrutinib-treated group, whereas the incidence of cardiovascular events and infections remained at a similar level between the two groups. These studies suggest that acalabrutinib may be a good alternative for patients intolerant to ibrutinib.

#### 3.1.3. Zanubrutinib

Zanubrutinib, a second-generation covalent BTK inhibitor, exhibits higher specificity and reduced off-target inhibition than ibrutinib [33]. A phase 2 study assessed its safety and efficacy among 91 CLL patients, with responses observed in 84.6% of them [34]. The phase 3 ALPINE study was designed to directly compare ibrutinib and zanubrutinib in patients with R/R CLL/SLL [35]. The study demonstrated the high efficacy of zanubrutinib. At 24 months, the PFS for zanubrutinib was 78.4% compared to 65.9% for ibrutinib [35]. Notably, increased PFS was also observed among patients with high-risk factors, such as TP53 mutation or 17p deletion. Importantly, the therapy with zanubrutinib was considered safer, with fewer adverse events recorded among patients receiving zanubrutinib compared to those on ibrutinib. Particularly, the incidence of atrial fibrillation/flutter was significantly lower in the zanubrutinib group compared to the ibrutinib group (5.2% vs. 13.3%).

#### 3.1.4. Orelabrutinib

As a selective BTK inhibitor, orelabrutinib represents an innovative small-molecule drug that irreversibly blocks BTK [36]. The orelabrutinib trial described in 2023 was designed to evaluate the efficacy and safety of orelabrutinib in patients with R/R CLL/SLL [37]. The study included 80 patients. The primary endpoint was CR rate, and secondary endpoints included PFS, OS, and safety [37]. Published data indicate that the drug is highly effective, with a 92.5% ORR and 23.1% CR. In addition, 86.8% of AEs were grade 1 or 2, suggesting that although highly effective, orelabrutinib has a favorable AE profile [37].

#### 3.1.5. Spebrutinib

Spebrutinib (CC-292) is an oral BTK inhibitor that uses covalent small-molecule binding to affect B-cell and Fc receptor signaling networks. In a Phase I study involving patients with CLL/SLL who received spebrutinib twice daily, the ORR was 53% [38]. The study was discontinued due to poor response.

#### 3.1.6. Tirabrutinib

Tirabrutinib shows greater BTK selectivity compared to ibrutinib [39]. Its therapeutic potential was evaluated in a multicenter phase 1 study that included 90 relapsed/refractory B-cell malignancy patients, with CLL cases comprising 25 of the participants [39]. Treatment response was achieved in all but one patient [39]. In a study by Munakata et al., the ORR was found to be 76.5% [40]. Adverse events occurred in all patients, with seven patients experiencing serious adverse reactions [40].

### 3.2. Non-Covalent BTKis

Unfortunately, treatment with covalent BTK inhibitors may lead to progressive disease, which is often associated with the previously mentioned C481S mutation. This mechanism of resistance has been addressed by the introduction of non-covalent BTK inhibitors, among which pirtobrutinib is, to date, the only one approved for the treatment of CLL patients who have relapsed on standard BTK inhibitor therapy.

Non-covalent BTKis have a different mode of action than their covalent ancestors. These compounds interact with the kinase by hydrogen, ionic bonds, and hydrophobic interactions [18]. Despite the greater number of interactions compared to covalent BTK inhibitors, their reversible and weaker nature results in a more favorable toxicity profile [18]. Interestingly, non-covalent BTK inhibitors have significantly longer half-lives, allowing for less frequent administration (Table 1).

#### 3.2.1. Pirtobrutinib

A multicenter study of pirtobrutinib was conducted in patients with previously treated B-cell malignancies. In Phase 1, the maximum tolerated dose was not reached, and no dose-limiting toxicities were reported [41]. The Phase 2 study was continued at a dose of 200 mg/day. Adverse events occurred in 10% of 323 patients, with neutropenia being the most common grade 3 or higher adverse event. Of note, no cases of atrial fibrillation or atrial flutter were reported during the study [41]. Pirtobrutinib shows activity in patients who progressed on prior BTK inhibitor therapy despite BTK mutation status [42]. Laboratory studies have shown that pirtobrutinib effectively suppresses BTK-dependent functions in both wild-type and C481S-mutant CLL cells [42]. Clinical samples from responding patients showed decreased BCR signaling and cell survival, although resistance developed over time. Genetic analysis identified novel BTK mutations as a potential resistance mechanism to non-covalent BTK inhibitors [42]. In the Shah et al. study, pirtobrutinib was used in patients who developed intolerance to at least one BTKi without progressive disease [43]. Of all patients, only 10.2% discontinued pirtobrutinib due to side effects. The overall response rate to pirtobrutinib was 76.9%. These results demonstrate that this drug may be a good alternative for patients who are intolerant to previous BTKi therapies [43]. Similarly, good efficacy results were shown in the study of Mato et al., where the ORR to pirtobrutinib was as high as 73.3% [8]. It is important to note that all patients had previously received treatment with covalent BTK inhibitors, and 40.5% were treated with both BTK inhibitors and BCL2 inhibitors. In the study, the PFS was 19.6 months. The most commonly seen AEs included infections, bleeding, and neutropenia; however, only 2.8% (nine patients) discontinued treatment due to AEs, indicating the overall safety of the therapy.

#### 3.2.2. Nemtabrutinib

Nemtabrutinib is an oral reversible inhibitor of BTK and the BTK mutant C481S. This compound is less selective than both pirtobrutinib and the second-generation covalent BTKis [18]. However, its off-target actions include the inhibition of PLCG2 by the downregulation of Lyn and Syk, which might be the reason for its good activity against CLL with *BTK*/*PLCG2* mutations [18,44]. The Phase I study MK-1026-001 evaluated nemtabrutinib in patients with R/R blood cancers who had received at least two prior therapies [45]. The study included 47 patients with CLL, non-Hodgkin’s lymphoma (NHL), and Waldenström’s Macroglobulinemia (WM). Up to 37 patients experienced grade 3 AEs. The ORR was 75%. This was the first phase of the study of nemtabrutinib in humans [45]. The BELLWAVE-011 study is a phase 3 trial assessing nemtabrutinib (65 mg) compared to ibrutinib (420 mg once daily) or acalabrutinib (100 mg twice daily) in previously untreated patients [46]. The study is supposed to include around 1200 participants, and as a marker of therapy efficacy, the safety of the treatment will be measured. The data from the trial are much awaited.

#### 3.2.3. Vecabrutinib and Fenebrutinib

Clinical trials of vecabrutinib were terminated due to suboptimal activity in CLL [47]. Sadly, greater dosages of the drug did not result in the clinical activity necessary to control the refractory disease [47]. However, there is some rationale for further evaluation, since vecabrutinib in combination with BCL2i has higher efficacy. The phase I study with fenebrutinib was terminated due to the following results in 15 patients: 8 patients experienced disease progression, 3 died, the clinician withdrew 2, 1 withdrew consent, and 1 discontinued due to a lack of support efficacy [48].

### 3.3. BTK Inhibitors Under Development

The development of novel BTK inhibitors has not ended, and novel compounds are under development. DTRMWXHS-12 is a novel chemical entity with a potent, selective, covalent inhibition of BTK, with desirable pharmacokinetic properties and efficacy in animal models [36]. In a recently published Phase 1a/1b study, DTRMWXHS-12 was well tolerated in patients with various R/R B-cell malignancies when co-administered with everolimus and pomalidomide [49]. In the study, 41.9% of the cohort achieved partial response (PR) or better, and two patients with DLBCL achieved complete response after triplet therapy. The phase 2 clinical trial is planned.

Another currently developed drug, luxeptinib (CG-806), is a first-in-class, non-covalent, and potent pan-FLT3/pan-BTK inhibitor [50]. In a Phase 1a/1b clinical trial, it showed a favorable toxicity profile as well as good clinical response among heavily pretreated acute myeloid leukemia (AML) patients [37]. What is more, luxeptinib was also tested in CLL/SLL patients, and its antitumor activity was observed even among patients previously exposed to ibrutinib. Currently, two active trials evaluate luxeptinib in AML and CLL (NCT03893682, NCT04477291).

## 4. BCL2 Inhibitors

The inner apoptosis pathway is strictly regulated by numerous mechanisms, such as the balance between pro-apoptotic and anti-apoptotic molecules [51,52]. The universal overexpression of the BCL-2 gene and the subsequent overproduction of the anti-apoptotic protein BCL-2 in cells of patients suffering from CLL disrupt this equilibrium [53,54]. Venetoclax (ABT-199) mimics a BH-3 molecule in order to bind to BCL-2, disrupts its signaling, and induces apoptotic pathway as a result [52]. Furthermore, it shows a greater affinity for BCL-2 than for BCL-W or BCL-XL, giving it a predominance over the ABT-737 and navitoclax (ABT-263) molecules, which both bind to BCL-W and BCL-XL, also present on platelets, thereby causing thrombocytopenia, which limits their clinical usefulness [52,55]. Due to ABT-737’s inability to be absorbed when taken orally, researchers developed navitoclax (ABT-263) as a modified version. While clinical trials showed that navitoclax was promising for treating hematologic cancers, its development was halted when researchers discovered it caused severe thrombocytopenia. This AE occurred because platelets need the protein Bcl-xL to survive [56].

Despite the excellent effect of initial treatment, we observe relapse in the majority of patients, partly due to resistance mechanisms [57]. The upregulation of BCL-2-related anti-apoptotic family members, changes in the BCL-2 gene that reduce venetoclax’s potential to bind, and abnormalities in genomic stability may be regarded as forms of resistance mechanisms. The overexpression of BCL-XL and MCL-1, observed in patients resistant to venotoclax, is one of the main causes of the mentioned struggle [58]. The potential cause of this impaired equilibrium may be activating alternative pathways and kinases such as BTK or phosphatidylinositol-3-kinase (PI3K) [59]. Moreover, the inhibition of these proteins appears to re-establish sensibility to venetoclax [60]. Complex karyotype and fludarabine-refractory CLL may be considered as significant risk factors of BCL2i treatment failure [61]. The third recognized resistance mechanism is the decreased possibility of venetoclax binding to the targeted spot caused by its mutations, for example, G101V mutation [62]. It is worth emphasizing that when mutations occur, they are only part of the resistance issue.

Following venetoclax’s success, several BCL2 inhibitors were developed or are currently under clinical evaluation. Sonrotoclax (BGB-11417) is a highly potent and selective second-generation BCL-2 inhibitor. BGB-11417 shows stronger binding affinity to the G101V mutant and is more effective against wild-type (WT) BCL-2 than venetoclax [63]. Moreover, BGB-11417 shows superior activity in inhibiting other mutants, i.e., D103Y [64]. Currently, clinical trials are being conducted to further evaluate BGB-11417 efficacy, tolerability, and safety as a monotherapy or combination treatment for CLL and other hematologic malignancies (NCT05479994, NCT04883957, NCT04277637).

Another novel, potent, oral, and specific inhibitor of BCL-2 is lisaftoclax (APG-2575), which was proven to surpass venetoclax. Deng et al. presented that treatment with lisaftoclax resulted in the increased activation of caspase 3/7 and enhanced apoptosis in both multiple myeloma and patient-derived primary samples [65]. The safety and efficacy of APG-2575 were investigated in a phase 1 trial in patients with R/R CLL/SLL and other hematologic malignancies (HMs). Notably, 14 of 22 patients with CLL/SLL had a PR according to the 2008 International Workshop on CLL criteria. The ORR for patients with CLL/SLL was 63.6%, although in patients with intolerance of BTKi treatment, the ORR was 80% [66].

LOXO-338 is another orally administered, novel BCL-2 inhibitor. Its preliminary efficacy was observed in a global, first-in-human phase 1 study, LOXO-BCL-20001, where LOXO-338 was administered orally to patients with advanced hematologic malignancies who had received standard therapy (NCT05024045). The preliminary results showed that, as of 18th October 2022, among the eight CLL/SLL patients, two had PR, three had stable disease (SD), and three were not evaluable [67].

## 5. BTK Inhibitor and BCL2 Inhibitor Synergism

Due to the recurrence of CLL observed in patients after treating them with one drug therapy (BTKi or BCL2i), there is a crucial need for new therapy strategies. Combining venetoclax with a BTK inhibitor appears to be a promising option due to various mechanisms of action and the ability to overcome single-agent refractoriness [68,69].

One of the pathways of BTKi-BCL-2i synergism is a change of dynamic between pro- and anti-apoptotic proteins. Deng et al. showed that ibrutinib increased BCL-2 dependence at the mitochondrial level, directly sensitizing CLL cells for venetoclax [70]. These substances complement each other when targeting pathologic factors on the tissue, cellular, and genetic levels. Ibrutinib preferentially targets and mobilizes the CLL cells in the lymph nodes, while venetoclax targets these in the bone marrow and blood [71]. Thus, the combination of both compounds targets two different subpopulations of CLL cells—dividing the subpopulation with ibrutinib and the resting one with venetoclax [72]. Moreover, ibrutinib, by decreasing the activity of CXCL12-CXCR4 signaling, results in an elevated efflux of malignant cells into the bloodstream, which allows better venetoclax effectivity due to the deprivation of prosurvival stimuli in the microenvironment [11,73]. Indeed, it has been demonstrated that the use of ibrutinib on CLL cells reduces the activity of both the BCR and NF-κB pathways [74]. It is important because CD-40-mediated NF-κB activation is one of the primary mechanisms of venetoclax resistance that occurs within the lymph nodes [75]. Ibrutinib treatment reduces the activation of this pathway, thereby leading to decreased Bcl-XL levels and subsequently to enhanced apoptosis sensitization [75]. Additionally, it decreases the expression of the CD-40 receptor, which attenuates cell survival even more [76]. Furthermore, BTK inhibition reduces MCL-1 levels, which is a protein from the BCL-2 family that is not targeted by BCL-2i [77]. Moreover, when combined with a BTK inhibitor, venetoclax causes additive cytotoxicity [71].

## 6. BTKi and BCL2i Combinations

Since both groups of drugs have proven effective in treating CLL, the rationale for combination therapy in treatment-naïve and R/R patients has emerged. To date, only ibrutinib and venetoclax therapy have been included in the current CLL management guidelines [5]. Other combinations of covalent and non-covalent BTK inhibitors with venetoclax or newer BCL2i group members are under evaluation. In this section, we examine recent advancements in combining BTKi and BCL2i compounds.

### 6.1. Ibrutinib + Venetoclax

In August 2022, the European Commission approved the combination of ibrutinib and venetoclax as a fixed-duration oral regimen for treating adult patients with previously untreated CLL. The approval was based on pivotal data from the phase 2 CAPTIVATE (NCT02910583) and phase 3 GLOW (NCT03462719) clinical trials. However, this treatment is not approved by the FDA, but the National Comprehensive Cancer Network Guidelines also support the use of this regimen in the United States.

The first phase 2 clinical trial to test the combination of venetoclax and ibrutinib was conducted at MD Anderson Cancer Center (NCT02756897). The study investigated the drug combination in two cohorts of CLL patients. Cohort I involved patients with R/R disease after at least one prior therapy. Cohort II included previously untreated patients with high-risk disease features, such as chromosome 17p deletion, mutated TP53, chromosome 11q deletions, unmutated IGHV, or age over 65 years. Treatment started with ibrutinib alone (420 mg daily, orally) for three cycles, followed by the addition of venetoclax with weekly dose escalation to 400 mg daily. The combination therapy was continued for 24 cycles. Patients with bone marrow (BM) undetectable minimal residual disease (uMRD) after 24 cycles discontinued both ibrutinib and venetoclax, while MRD-positive patients could continue ibrutinib. In cohort I, after three cycles of ibrutinib monotherapy, none of the 73 evaluable patients achieved BM uMRD. After 12 cycles of the combination therapy, 29 out of 60 patients (48%) achieved BM uMRD remission. After 24 cycles, 16 out of 24 patients (67%) achieved BM uMRD remission. Median OS and PFS were not reached by month 36. Grade 3/4 neutropenia occurred in 29% of patients and grade 3/4 thrombocytopenia in 3%. Atrial fibrillation was reported in 9% [78]. In the cohort of untreated high-risk and older patients with CLL, 80 patients were enrolled. The median follow-up was 38.5 months. After 12 cycles of combination therapy, 56% of patients achieved BM uMRD remission, increasing to 66% after 24 cycles. At both 12 and 24 cycles, 69% of patients achieved a CR or CRi. The estimated 3-year PFS rate and OS were 93% and 96%, respectively. Grade ≥ 3 hematologic AEs included neutropenia in 51% of patients and thrombocytopenia in 2%. The most common non-hematologic grade ≥ 3 AEs were atrial fibrillation (10%) and hypertension (10%) [79]. At the 5-year follow-up, 72% of patients achieved BM uMRD as their best response. The 5-year PFS rate was 90.1%, and the 5-year OS rate was 95.6% [80].

The use of this drug combination in previously untreated patients was further explored in the CAPTIVE trial. The multicenter, two-cohort phase 2 study evaluated two approaches to treatment with ibrutinib and venetoclax in treatment-naïve patients with CLL under the age of 70. The trial assessed both MRD-guided treatment discontinuation and fixed-duration (FD) therapy. A total of 164 patients were enrolled, with a median follow-up of 31.3 months. Treatment began with oral ibrutinib (420 mg once daily) as a single agent for three cycles, followed by the addition of venetoclax (target dose of 400 mg once daily after, reached after a standard 5-week ramp-up) for 12 cycles. After completing 12 cycles of combination therapy, patients in the MRD-guided cohort were randomized based on their MRD status. Those with confirmed uMRD were assigned to receive either a double-blind placebo (n = 43) or ibrutinib (n = 43). Patients with unconfirmed uMRD were randomized to receive either single-agent ibrutinib (n = 31) or continued ibrutinib plus venetoclax (n = 32). The primary endpoint was the 1-year disease-free survival (DFS) rate, comparing placebo to ibrutinib in the confirmed uMRD population. Secondary endpoints included response rates, uMRD status, and safety. In the pre-randomization phase, 75% (123 of 164) of all treated patients achieved the best MRD response of uMRD in peripheral blood (PB), while 68% (112 of 164) achieved uMRD in BM. Of the 149 patients eligible for random assignment, 86 met the confirmed uMRD criteria, while the remaining 63 were classified as having unconfirmed uMRD. For patients with confirmed uMRD, the 1-year DFS rate was 100% for those who received ibrutinib, compared to 95% for those who received placebo. However, this difference was not statistically significant (*p* = 0.15). In patients with unconfirmed uMRD, the estimated 30-month PFS rates were 95% with ibrutinib alone and 97% with ibrutinib plus venetoclax. The safety profile of the combination of ibrutinib and venetoclax was consistent with the known AEs of each drug individually, with no new safety signals observed. The high 1-year DFS rate in confirmed uMRD patients receiving a placebo following 12 cycles of the ibrutinib and venetoclax combination therapy, as well as the high 30-month PFS rates of the MRD-positive population after this treatment suggest the significant potential of FD treatment with this drug combination using MRD guidance [81]. In the FD cohort (N = 159), with a median follow-up of 27.9 months, the ibrutinib plus venetoclax combination achieved deep and durable responses. The CR rate was 56%, and the best uMRD rates were 77% in PB and 60% in BM. The most common grade ≥ 3 AEs were neutropenia (33%) and hypertension (3%) [82]. At the long-term follow-up of 61.2 months in the FD cohort, 5-year PFS and OS rates were 67% and 96%, respectively. Patients with uMRD at 3 months post-treatment had higher 5-year PFS rates than those without uMRD: 83% vs. 48% in PB, and 84% vs. 50% in BM [83].

While the CAPTIVATE trial focused on younger patients, the phase 3 GLOW study evaluated efficacy and safety in elderly or unfit populations. The study included patients aged ≥65 years, or 18–64 years with a Cumulative Illness Rating Scale score > 6. A total of 211 treatment-naïve patients were enrolled and randomly assigned to 12 cycles of FD ibrutinib–venetoclax (N = 106) or to 6 cycles of chlorambucil–obinutuzumab (N = 105). After a median follow-up of 27.7 months, PFS was significantly longer in the ibrutinib–venetoclax group compared to the chlorambucil–obinutuzumab group (hazard ratio, 0.216). Median PFS was not reached in the ibrutinib–venetoclax arm, while it was 21 months in the chlorambucil–obinutuzumab arm. At 3 months post-treatment, uMRD rates were higher in the ibrutinib–venetoclax arm compared to the chlorambucil–obinutuzumab arm in both BM (52% vs. 17%) and PB (55% vs. 39%). At the 1-year follow-up, 85% of patients treated with ibrutinib–venetoclax who achieved uMRD maintained their status. Additionally, the ibrutinib–venetoclax combination demonstrated a higher CR/CRi rate than chlorambucil plus obinutuzumab (39% vs. 11%). The safety profile of the ibrutinib plus venetoclax combination was consistent with the known safety profiles of the drugs individually. The most common grade ≥ 3 AEs were neutropenia (35%), diarrhea (10%), and hypertension (10%). The once-daily FD combination of ibrutinib and venetoclax led to deeper and more durable responses than chlorambucil–obinutuzumab and improved PFS in older, comorbid, treatment-naïve CLL patients [84]. At a long-term median follow-up of 46 months, PFS remained higher in the ibrutinib–venetoclax arm compared to the chlorambucil–obinutuzumab arm. The median PFS was still not reached in the ibrutinib–venetoclax arm. At 36 months, CR/Cri rates and estimated 42-month OS were significantly higher in the ibrutinib–venetoclax arm compared to the chlorambucil–obinutuzumab arm (93% vs. 60% and 87.5% vs. 77.6%, respectively) [85]. The latest data from the GLOW study, following 64 months of follow-up, have recently been published and continue to show the favorable profile of ibrutinib and venetoclax compared to chlorambucil and obinutuzumab [86]. At 60 months, the PFS and OS were 59.9% and 81.6% for the ibrutinib and venetoclax combination, compared to 17.8% and 60.8% in the chlorambucil and obinutuzumab group. Notably, prolonged PFS occurred regardless of IGHV mutation status. Additionally, the toxicity analysis indicated a relatively longer emergence of grade 3/4 AEs within the ibrutinib and venetoclax group. However, the overall treatment-emergent AE (TEAE)-free PFS was favorable for the BTKi/BCL2i combination when compared to the chemoimmunotherapy group.

In the CLARITY trial, therapy duration was determined by the dynamics of MRD negativity status. Patients achieving uMRD in both PB and BM within six months of combination therapy stopped ibrutinib–venetoclax treatment after 12 cycles. Those who reached uMRD between cycles 12 and 24 discontinued the combination therapy at the 24th cycle. Meanwhile, patients with detectable MRD at the final evaluation transitioned to ongoing ibrutinib monotherapy. In a phase 2 study, ibrutinib was combined with venetoclax in 50 patients with R/R CLL. After two months of ibrutinib monotherapy, venetoclax was added, escalating to a final daily dose of 400 mg. The primary endpoint was achieving uMRD after 12 months of therapy. Fourteen of the fifty evaluable participants achieved MRD negativity in PB and BM after 6 months of combination therapy [87,88]. MRD negativity continued to improve over time: 20 of 50 patients (40%) achieved MRD negativity by month 14, and 24 of 50 (48%) achieved it by month 26. Notably, 17 out of 23 patients who discontinued treatment before 36 months did so because they achieved uMRD. At 36 months, 14 out of 18 patients who stopped therapy due to uMRD maintained a negative status. The initial rate of CLL depletion was predictive of longer-term treatment response. Patients with persistent MRD after 12 months of combination therapy typically experienced a slow decline in disease levels, like that observed with ibrutinib monotherapy. Regarding AEs, one case of tumor lysis syndrome was observed. Other AEs were manageable, with the most common being grade 3/4 neutropenia [89].

Following the positive results from the CLARITY trial, the UK CLL group investigated whether ibrutinib–venetoclax combination therapy with MRD-guided duration is more effective than fludarabine–cyclophosphamide–rituximab (FCR). In the phase 3, multicenter FLAIR trial, 523 patients were randomly assigned to the ibrutinib–venetoclax group (N = 260) or the FCR group (N = 263). After a median follow-up of 43.7 months, the rate of disease progression or death was lower in the ibrutinib–venetoclax group compared to the FCR group (4.6% and 28.5%, respectively). The estimated 3-year PFS was higher with ibrutinib–venetoclax than with FCR (97.2% vs. 76.8%). The 3-year OS was also higher with ibrutinib–venetoclax than with FCR (98.0% vs. 93.0%). uMRD in BM at any time was observed in 61.9% of patients in the ibrutinib–venetoclax group compared to 40.3% in the FCR group. The MRD-guided treatment duration with ibrutinib and venetoclax resulted in better OS and PFS among patients with previously untreated CLL. Notably, ibrutinib–venetoclax showed superiority compared to FCR in higher-risk IGHV-unmutated patients [89].

The SAKK 34/17 trial evaluated the efficacy of a prolonged 24-month induction phase of ibrutinib–venetoclax with a longer ibrutinib lead-in period of six cycles to enhance uMRD and CR/CRi rates. The primary results showed that after 24 cycles of therapy, 40% of patients achieved uMRD, with CR/CRi exceeding the outcomes obtained in the CAPTIVATE and CLARITY studies [81,87]. The results support the idea that prolonging therapy duration can enhance therapeutic efficacy [90].

Several other clinical trials exploring the ibrutinib–venetoclax combination are ongoing. The phase 3 CLL17 trial led by the German CLL Study Group is comparing FD venetoclax–ibrutinib combination therapy for 15 cycles with FC ibrutinib–obinutuzumab combination therapy for 12 cycles, as well as continuous ibrutinib therapy until progression (NCT04608318). The phase 2 VALUABLE trial aims to evaluate the benefits of adding ibrutinib to 12 months of venetoclax (administered as a single agent for 6 months, followed by a combination with rituximab for the next 6 months) in treatment-naïve CLL patients, based on an MRD-guided approach.

There are benefits of combining ibrutinib with venetoclax as a treatment strategy for CLL patients in treatment-naïve and R/R settings. However, there is still no clear agreement on the optimal treatment duration, and many trials apply MRD-guided treatment discontinuation. Upcoming data from ongoing trials may help optimize ibrutinib–venetoclax combination therapy and explore new treatment approaches involving BTKi and BCL2i combinations. The results of major clinical trials evaluating ibrutinib + venetoclax combination for patients with CLL are presented in Table 2.

### 6.2. Acalabrutinib + Venetoclax

AMPLIFY, the first phase 3 randomized clinical trial, compared the efficacy and safety of acalabrutinib and venetoclax combination with and without obinutuzumab in fixed-duration therapy versus standard-of-care chemoimmunotherapy in previously untreated adult CLL patients without del(17p) and/or *TP53* mutations. The study randomized patients 1:1:1 to receive acalabrutinib plus venetoclax (AV), acalabrutinib plus venetoclax with obinutuzumab (AVO), or standard-of-care chemoimmunotherapy. The AV and AVO arms were administered for a fixed duration of 14 cycles (each 28 days), while the control arm received 6 cycles of chemoimmunotherapy. Both trial arms showed durable responses, with estimated 36-month PFS rates of 76.5%, 83.1%, and 66.5% for AV, AVO, and chemoimmunotherapy, respectively (Table 3). ORR was significantly higher in the AV and AVO groups (acalabrutinib with venetoclax: 92.8%; with added obinutuzumab: 92.7%) compared to chemoimmunotherapy (75.2%). Interim OS data demonstrated a statistically significant advantage favoring acalabrutinib–venetoclax. The safety profile was consistent with the previous observations. In AMPLIFY, the acalabrutinib and venetoclax combination demonstrated favorable tolerability with low incidences of AEs: atrial fibrillation (all grades: 0.7%; grade ≥ 3 0.3%), grade ≥ 3 hypertension (2.1%), and grade ≥ 3 bleeding (1%). What is more, in the AV arm, a low rate of tumor lysis syndrome (TLS) (0.3% of any severity) and treatment discontinuation due to adverse events (7.9%) was observed [92].

While the AMPLIFY trial established fixed-duration AVO as a new standard of care in treatment-naive CLL, it excluded patients with high-risk *TP53* aberrations. Another phase II study enrolled 72 patients, including 45 with TP53 aberrations [94]. MRD-guided AVO therapy duration demonstrated high activity across all genetic subgroups, with high-risk TP53-aberrant patients achieving a 42% rate of BM-uMRD with CR at the start of cycle 16 and 80% achieving the best BM-uMRD rate. Response durability was evidenced by 70% 4-year PFS, with no CLL progression during initial therapy (Table 3).

The CLL2-BAAG phase II trial [93] investigated AVO in relapsed/refractory CLL (Table 3). The study population included patients who had received BTKi and/or venetoclax (40%) and those with TP53 aberrations (31.8%), with a median of one prior therapy. After a median follow-up of 36.3 months and an off-treatment duration of 21.9 months, uMRD in peripheral blood was achieved in 93.3% of patients, including 94.4% of BTKi/venetoclax-exposed and 92.9% of TP53-aberrant cases. The estimated 3-year PFS and OS rates were 85% and 93.8%, respectively.

The promising efficacy of acalabrutinib–venetoclax combinations has prompted numerous ongoing clinical trials across various therapeutic settings (Appendix A).

### 6.3. Zanubrutinib + Venetoclax

The good tolerability and efficacy of zanubrutinib opened the possibility for combination treatments with other drugs, among them venetoclax. This combination is promising, with the results of cell line studies showing significant synergy between the two agents [100]. Indeed, venetoclax and zanubrutinib co-therapy was an arm D of the SEQUOIA clinical trial, which cemented zanubrutinib’s place in the modern therapy of CLL (Table 3). In the arm, treatment-naïve CLL/SLL patients received three months of zanubrutinib treatment followed by up to 24 cycles of combination therapy with venetoclax until progression, uMRD achievement, or intolerable toxicity was reached [95]. Of note, only del(17p) patients were eligible for this treatment. Adverse events were reported in 82.9% of the patients; however, serious AEs were observed only among four patients (11.4%). The first efficiency analysis is promising, with an ORR of 96.8% and only one patient progressing during the treatment. It is even more outstanding taking into consideration the fact that 94.3% of the cohort had high-risk characteristics of the disease, such as Binet C stage, bulky disease ≥ 5 cm, or unmutated IGHV. The complete study results have not been published yet. Similarly, good tolerability was also seen in a regimen combining zanubrutinib, venetoclax, and obinutuzumab in previously untreated CLL/SLL patients [96]. Importantly, uMRD was observed in 33 out of 37 patients (89%) after a median follow-up of 25.8 months. Moreover, MRD negativity persisted at 94% after a median of 15.8 months. However, some researchers postulate adding obinutuzumab only in patients who do not reach uMRD after zanubrutinib and venetoclax [101]. This approach will surely decrease monoclonal-antibody-related toxicities. Several ongoing clinical trials are investigating the safety and tolerability of the combination of zanubrutinib and venetoclax in various clinical scenarios (Appendix A). Lastly, there is an intriguing new way of administering both drugs in one long-lasting injection [102]. In cell line models, therapy with zanubrutinib–venetoclax nanoformulation showed increased uptake when compared to the free drugs. Additionally, mice models showed an increased half-life of both drugs: a 43-fold increase in venetoclax and a 5-fold increase in zanubrutinib, which may translate into better synchronization, longer exposure, and better clinical efficiency. On the other hand, injections may cause lower compliance; thus, research needs to continue to produce stable formulas in pill form.

### 6.4. Orelabrutinib + Venetoclax

Sadly, to this date, there are no trials assessing the use of orelabrutinib and venetoclax in the clinic. However, there is some preclinical evidence suggesting that both drugs work synergistically to induce cell death [103]. In double-hit lymphoma cells, the combination of both drugs resulted in decreased levels of proliferation, increased activation of the intrinsic apoptosis pathway, and increased cell cycle arrest. Additionally, the synergism of both medications was observed to be the result of interference in the PI3K/AKT and p38/MAPK signaling pathways. However, additional data from mice models and CLL cell lines are scarce; thus, this combination needs further evaluation.

### 6.5. Pirtobrutinib + Venetoclax

The first non-covalent BTKi grabbed early attention as a possible combination choice with venetoclax, since both drugs might be used after first-line covalent BTKi therapy failure. The results of the Phase 1b BRUIN study investigated the efficacy and safety of both drugs with or without rituximab in CLL patients [97] (Table 3). Seventeen patients (68%, N = 25) of the cohort had had prior therapy with covalent BTKis. In the study, ten patients received pirtobrutinib and venetoclax alone (PV), and fifteen patients took both drugs with an addition of rituximab (PVR). At the data cutoff, treatment was discontinued in 22 patients, among which 14 received a total of 24 cycles and 8 discontinued the therapy. The therapy is considered safe, with the most common treatment-related AEs of grade 3 or higher being neutropenia, infections, diarrhea, and thrombocytopenia. The efficacy analysis revealed an ORR of 96% among all treated patients. Ten patients (40%) achieved CR, while PR was observed in fourteen patients (56%). Surprisingly, CRs occurred more often in the PV group (46.7%) than in the PVR arm (30.0%). Importantly, all patients previously treated with covalent BTKis responded to the treatment, and 8 (47%) out of 17 patients achieved CR. uMRD was seen in 85.7% of patients receiving PV and 90.0% of patients receiving PVR.

The combination treatment with pirtobrutinib and venetoclax is also currently being examined in treatment-naïve patients with the addition of obinutuzumab [104]. The enrolled patients received 200 mg of pirtobrutinib daily, with a standard six cycles of obinutuzumab, and a standard venetoclax ramp-up until 400 mg was reached. The study continued for 13 cycles; however, the patients who did not achieve uMRD could continue PV for an additional 12 cycles. At the end of the seventh cycle, 65% (28/43) and 79% (34/43) achieved uMRD (10^−6^ sensitivity) in the bone marrow and blood, respectively. The numbers rose to 81% (22/27) and 89% (24/27) at the end of the 13th cycle among patients who reached this time point. The ongoing clinical trials assessing these drug combinations are listed in Appendix A.

### 6.6. Nemtabrutinib + Venetoclax

There is a preclinical rationale for the combination of nemtabrutinib and venetoclax [105]. The addition of nemtabrutinib to venetoclax in silico increases the cytotoxicity by 10% (6% when ibrutinib is added). Furthermore, nemtabrutinib-treated patients’ cells are dependent on BCL2 and BCL-xL, which suggests the probable clinical benefit of venetoclax introduction in this group. In addition, an in vivo study showed that the combination of both drugs significantly prolonged survival among mice grafted with splenocytes of a single donor in comparison to ibrutinib, nemtabrutinib, or ibrutinib and venetoclax. Considering the good preclinical results, a study evaluating the combination of nemtabrutinib and venetoclax vs. venetoclax and rituximab is currently active (NCT05947851). The researchers will investigate whether nemtabrutinib and venetoclax are superior to the latter combined with rituximab in patients with R/R CLL. The results of the study are much awaited.

### 6.7. Sonrotoclax Combinations

Venetoclax’s success in treating hematological malignancies is the reason for the continuous research and widening of this group of BCL2 inhibitors. One of the new members of the group is sonrotoclax, which was preliminary tested with zanubrutinib as a possible treatment option for CLL/SLL. Intriguingly, the combination treatment showed a good tolerability profile with high anti-tumor efficiency [106]. Of note, the combination therapy with zanubrutinib achieved higher levels of PR, and was better than sonrotoclax alone (72.7% vs. 66%). Importantly, the ORR among patients receiving the combination therapy was 100%, and the rate of CR increased over time [98] (Table 3). In a group receiving a lower dose (160 mg) of sonrotoclax, uMRD was achieved in 50% at 24 weeks and 73% at 48 weeks of follow-up. The higher dose of sonrotoclax was even more efficient, with 65% of the evaluated patients achieving uMRD at week 24, and a staggering 100% of patients were uMRD-positive towards 48 weeks of the study. The results are promising; thus, several clinical trials will investigate the possibility of introducing the sonrotoclax–zanubrutinib combination into standard CLL patient care (Appendix A). Moreover, the combination is being investigated among patients with NHL, WM, and mantle cell lymphoma [107]. To this date, no clinical data have been presented involving other BTKi and sonrotoclax combinations; however, with increased use of the second generation of BCL2is, these data should arrive soon.

### 6.8. Lisaftoclax Combinations

Another new member of the BCL2 inhibitors group is lisaftoclax. Its combination with acalabrutinib or rituximab was investigated in a phase 1b/2 study in patients with treatment-naïve, relapsed/refractory, or prior ven-treated CLL/SLL [99] (Table 3). A combination of lisaftoclax and acalabrutinib resulted in ORR 96.6%, with 86% 18-month PFS. In a ven-exposed group, ORR was 85.7%. This figure was 100% in the ven-exposed but BTKi-naïve group, and 66.7% in ven- and BTKi-exposed patients, with a 73% 18-month PFS rate. These data suggest that lisaftoclax and acalabrutinib combination might be a promising treatment option for patients with prior venetoclax exposure, including those with progression on venetoclax. No drug–drug interactions (DDIs) of lisaftoclax with acalabrutinib/rituximab were found, and no discontinuations were attributed to lisaftoclax treatment-related adverse events (TRAEs). The study’s authors will further evaluate the lisaftoclax and acalabrutinib combination in patients with prior venetoclax exposure in a global phase 3 clinical study, GLORA. This study and other clinical trials containing lisaftoclax and BTKi combinations are presented in Appendix A. Moreover, lisaftoclax alone or combined with ibrutinib or rituximab is investigated in treating WM [108].

## 7. Conclusions

In conclusion, recent years have seen some of the biggest advancements in CLL treatment. One of the hopes of modern-day leukemia management is the combination of BTKi with BCL2i. To date, only the venetoclax and ibrutinib regimen is listed in the 2024 ESMO recommendations. However, the combination therapies of venetoclax with acalabrutinib, zanubrutinib, and pirtobrutinib show much promise, with potential for better efficacy and more acceptable toxicity profiles. There are, however, not enough data to draw conclusions about newer compounds, but the early results are encouraging. The additive value of monoclonal antibodies is also being investigated. Thus, the probable superiority of triple-drug therapy over BTKis and BCL2is will soon be known.

## Figures and Tables

**Figure 1 ijms-26-03193-f001:**
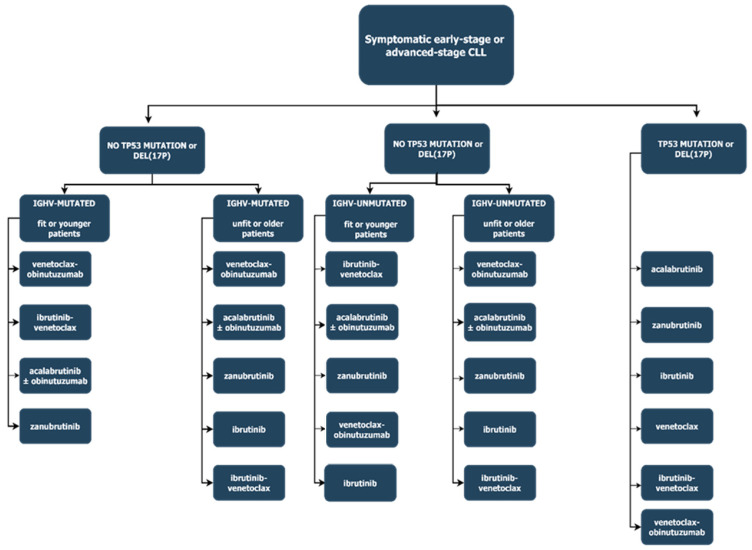
First-line treatment strategies for symptomatic chronic lymphocytic leukemia. Modified from Eichhorst et al. [5].

**Table 1 ijms-26-03193-t001:** BTKi comparison.

Drug	Ibrutinib	Acalabrutinib	Zanubrutinib	Orelabrutinib	Pirtobrutinib	Nemtabrutinib	Ref.
Class	First-generation covalent BTKi	Second-generation covalent BTKi	Second-generation covalent BTKi	Second-generation covalent BTKi	Non-covalent BTKi	Non-covalent BTKi	[11,16,17,18]
Binding with BTK	Irreversible at C481S	Irreversible at C481S	Irreversible at C481S	Irreversible at C481S	Reversible at the ATP-binding site	Reversible at E475 and Y476 residues of BTK	
Specificity	Low	High	High	High	Very High	Low	
Off-targets	ITK, EGFR, CSK, ErbB2, and TEC	Reduced off-target effects	Reduced off-target effects, TEC inhibition	Reduced off-target effects	Reduced off-target effects	SRC, AKT, and ERK inhibition	
Half-life (hours)	4–13 h	1–2 h	2–4 h	1.5–4 h	20 h	20–30 h	

**Table 2 ijms-26-03193-t002:** Clinical trials evaluating ibrutinib and venetoclax combination for CLL patients.

Clinical Trial	Phase	No. of Participants	%ORR	CR/CRi Rate	uMRD Rate	PFS Rate	OS Rate	Ref.
MD Anderson Cancer Center—pts. with R/R CLL	2	80	N/A	N/A	48% at cycle 12 (BM), 67% at cycle 24 (BM)	c. 75% at 3 years	Over 90% at 3 years	[78]
MD Anderson Cancer Center—treatment-naïve high-risk pts.	2	120	N/A	69% at cycle 12 and 24	56% at 12 cycles (BM), 66% at 24 cycles (BM)	90.1% at 5 years	95.6% at 5 years	[79,80]
CAPTIVATE—FD cohort	2	159	96%	56%	77% (PB) and 60% (BM) at 12 cycles	67% at 5 years	96% at 5 years	[82,83]
CAPTIVATE—MRD cohort	2	164	97%	46%	75% (PB) and 68% (BM) at 12 cycles	≥95% at 2.5 years	99% at 3 years	[81]
GLOW	3	106	N/A	43%	55% at 12 cycles (PB)	74.6% at 3.5 years	87.5% at 3.5 years	[85]
CLARITY	2	50	89%	51%	40% at 12 cycles (BM), 48% at 24 cycles (BM)	98% at a median follow-up of 21.1 months	100% at a median follow-up of 21.1 months	[87,89]
FLAIR	3	260	86.5% at 9 months	59.2%	52.4% at 2 years (BM), 65.9% at 5 years (BM)	97.2% at 3 years	98.0% at 3 years	[91]

BM—bone marrow; CR—complete remission; CRi—complete remission with incomplete count recovery; MRD—measurable residual disease; N/A—not available; OS—overall survival; ORR—overall response rate; PB—peripheral blood; PFS—progression-free survival; R/R CLL—relapsed and refractory chronic lymphocytic leukemia.

**Table 3 ijms-26-03193-t003:** Preliminary data from clinical trials evaluating BTKi and BCL2i combinations other than venetoclax/ibrutinib.

Clinical Trial	Phase	Combination	No. of Participants	%ORR	CR/CRi Rate	uMRD Rate	PFS Rate	OS Rate	Ref.
AMPLIFY	3	Acalabrutinib + venetoclax +/− obinutuzumab	867	AV 92.8%, AVO 92.7%	N/A	AV 34.4% at cycle 14, day 28; AVO 67.1% at cycle 6, day 1	AV 76.5%, AVO 83.1% at 36 months	AV 94.1%, AVO 87.7% at 36 months	[92]
CLL2-BAAG	2	Acalabrutinib + venetoclax + obinutuzumab	46	N/A	N/A	93.3% at 36.3 months	85% at 3 years	93.8% at 3 years	[93]
Davids et al.	2	Acalabrutinib + venetoclax	72	Best ORR: Patients with *TP53* aberration 98% and all-comers 99%	Patients with *TP53* aberration 48.9% and all-comers 47.2% at cycle 16	BM-uMRD: Patients with *TP53* aberration 71% and all-comers 78% at cycle 16	70% at 4 years	Patients with *TP53* aberration 88% and all-comers 100% at 4 years	[94]
SEQUOIA (Arm D)	3	Zanubrutinib + venetoclax	35	96.8%	11%	N/A	N/A	N/A	[95]
Soumerai et al.	2	Zanubrutinib + venetoclax + obinutuzumab	39	100%	57%	89%	Not reached	N/A	[96]
BRUIN	1b	Pirtobrutinib + venetoclax (PV) +/− rituximab (PVR)	25	PV 93.3% PVR 100%	40%	PV 85.7% PVR 90%	PV 79.6% at 24 months PVR 80% at 18 months	PV 92.9% at 24 months PVR 80% at 18 months	[97]
Tam et al.	1/2	Zanubrutinib + sonrotoclax	94	100%	N/A	75% at week 48	N/A	N/A	[98]
Davids et al.	1b/2	Lisaftoclax + acalabrutinib (LA) or rituximab (LAR)	176	Lisaftoclax + acalabrutinib 96.6%	N/A	N/A	86% at 18 months	N/A	[99]

BM—bone marrow; CR—complete remission; CRi—complete remission with incomplete count recovery; MRD—measurable residual disease; N/A—not available; OS—overall survival; ORR—overall response rate; PFS—progression-free survival.

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
