# Peer review of "Double Strike in Chronic Lymphocytic Leukemia—The Combination of BTK and BCL2 Inhibitors in Actual and Future Clinical Practice"

_ijms, 2025, doi:10.3390/ijms26073193_

Round 1
Reviewer 1 Report
Comments and Suggestions for Authors
The paper present a good review no the use of inhibitors to fight CLL. No major issues with the with the exception of one error listed below that will need to be corrected. The work is well presented and the different inhibitors and clinical trials well summarized.
Issue: in page 6, in the section on BCL2 inhibitors. In the second sentence, the authors mistakenly describe BCL2 as a pro-apoptotic proteins/genes, but it is the opposite. This can be fixed by changing in the second sentence in section 4. BCL2 inhibitors "overproduction of the pro-apoptotic..." to "overproduction of the anti-apoptotic...".
Author Response
The paper present a good review no the use of inhibitors to fight CLL. No major issues with the with the exception of one error listed below that will need to be corrected. The work is well presented and the different inhibitors and clinical trials well summarized.
Issue: in page 6, in the section on BCL2 inhibitors. In the second sentence, the authors mistakenly describe BCL2 as a pro-apoptotic proteins/genes, but it is the opposite. This can be fixed by changing in the second sentence in section 4. BCL2 inhibitors "overproduction of the pro-apoptotic..." to "overproduction of the anti-apoptotic...".
Response:
Thank you very much for the kind words and comments regarding our paper. We addressed the issue raised by you accordingly. We hope that our article is now suitable for the publication.
Reviewer 2 Report
Comments and Suggestions for Authors
Dear Authors, I would like to congratulate you on your work. I have some minor suggestions, as well as a few major ones. Beginning with the more important aspects affecting the quality of the work:
- The paragraph explaining the additive effect of combined BTK and BCL2 inhibition is overly abbreviated. Several aspects, such as the involvement of both BTK and BCL2 in pathways like NF-κB, are not mentioned at all.
- Distinguishing between covalent and non-covalent BTK inhibitors goes beyond just the type of bond, despite the name. The pharmacokinetic consequences of these different modes of action were not addressed by the authors.
- In general, topics related to the structure, pharmacokinetics of the drugs, and the molecular aspects of the inhibitors' mode of action are barely addressed or largely ignored.
The conclusion section, despite its length, lacks substance.
Minor concerns:
- The text would benefit from thorough proofreading, as some sentences are overly convoluted and difficult for the reader to understand.
- Several sentences introduce vital information yet lack proper citation.
- Abbreviations are chaotic, introduced several times or introduced after using the words for a couple of times e.g. Waldeström's Macroglobulinemia.
Author Response
Dear Authors, I would like to congratulate you on your work. I have some minor suggestions, as well as a few major ones. Beginning with the more important aspects affecting the quality of the work:
- The paragraph explaining the additive effect of combined BTK and BCL2 inhibition is overly abbreviated. Several aspects, such as the involvement of both BTK and BCL2 in pathways like NF-κB, are not mentioned at all.
- Distinguishing between covalent and non-covalent BTK inhibitors goes beyond just the type of bond, despite the name. The pharmacokinetic consequences of these different modes of action were not addressed by the authors.
- In general, topics related to the structure, pharmacokinetics of the drugs, and the molecular aspects of the inhibitors' mode of action are barely addressed or largely ignored.
The conclusion section, despite its length, lacks substance.
Minor concerns:
- The text would benefit from thorough proofreading, as some sentences are overly convoluted and difficult for the reader to understand.
- Several sentences introduce vital information yet lack proper citation.
- Abbreviations are chaotic, introduced several times or introduced after using the words for a couple of times e.g. Waldeström's Macroglobulinemia.
Response:
Thank you very much for your review and most useful suggestions. We have updated the manuscript accordingly.
Major concerns:
- We have included a paragraph discussing the added value of ibrutinib in NF-κB-mediated resistance to venetoclax.
- We have updated the manuscript to include an additional paragraph in the non-covalent BTKi section. Furthermore, we have added further data regarding nemtabrutinib in its section.
- We have prepared an additional table that illustrates the comparison of several relevant features of BTKi.
- The conclusion section was shortened and rewritten to be more informative.
Minor concerns:
- We have read the manuscript again and corrected some of the grammatical errors. We hope that the manuscript is now readable and ready to be published.
- We have read the manuscript again and added some of the missing citations.
- We unified and corrected the abbreviations.
Reviewer 3 Report
Comments and Suggestions for Authors
The manuscript by Zygmunciak et al. entitled “Double Strike in Chronic Lymphocytic Leukemia - The Combination of BTK and BCL2 Inhibitors in Actual and Future Clinical Practice” reviews the current progress in the development and clinical implication of novel drugs for Chronic Lymphocytic Leukemia (CLL). The authors present a comprehensive review of the relevant literature and the clinical trials in clear, concise and informative way.
First they point out the factors for worst prognosis in CLL - TP53 mutations , detectable del(17p), unmutated immunoglobulin heavy-chain variable region gene (IGHV) status and then describe the available treatments based on Bruton’s Tyrosine Kinase inhibitors and BCL2 inhibitors. They discuss monotherapies as well as different drug combinations with detailed analysis of the observed clinical outcomes.
The authors refer to 97 citations and present 2 tables (plus one supplementary table) with clinical trials results.
The text is well written; the results are clearly summarized and presented.
Taking into account the importance of the discussed topic and the high quality of the manuscript, I recommend publication in IJMS.
I have 2 minor notes to the authors, that can be corrected during the proof stage:
- There is 1 unnecessary comment left in the Suplementary table.
- In the abstract - use “Bruton's Tyrosine Kinase Inhibitors (BTKIs)” instead of “Bruton kinase inhibitors”
Author Response
The manuscript by Zygmunciak et al. entitled “Double Strike in Chronic Lymphocytic Leukemia - The Combination of BTK and BCL2 Inhibitors in Actual and Future Clinical Practice” reviews the current progress in the development and clinical implication of novel drugs for Chronic Lymphocytic Leukemia (CLL). The authors present a comprehensive review of the relevant literature and the clinical trials in clear, concise and informative way.
First they point out the factors for worst prognosis in CLL - TP53 mutations, detectable del(17p), unmutated immunoglobulin heavy-chain variable region gene (IGHV) status and then describe the available treatments based on Bruton’s Tyrosine Kinase inhibitors and BCL2 inhibitors. They discuss monotherapies as well as different drug combinations with detailed analysis of the observed clinical outcomes.
The authors refer to 97 citations and present 2 tables (plus one supplementary table) with clinical trials results.
The text is well written; the results are clearly summarized and presented.
Taking into account the importance of the discussed topic and the high quality of the manuscript, I recommend publication in IJMS.
I have 2 minor notes to the authors, that can be corrected during the proof stage:
- There is 1 unnecessary comment left in the Suplementary table.
- In the abstract - use “Bruton's Tyrosine Kinase Inhibitors (BTKIs)” instead of “Bruton kinase inhibitors”
Response:
Thank you kindly for your thoughtful review of our work. We have revised the manuscript in accordance with your suggestions. We hope that the article is now ready for publication.